# The Impact of the Gender Digital Divide on Sustainable Development: Comparative Analysis between the European Union and the Maghreb

**Hayet Kerras** * , **Jorge Luis Sánchez-Navarro** , **Erasmo Isidro López-Becerra** and **María Dolores de-Miguel Gómez** *

Departamento de Economía de la Empresa, Escuela Técnica Superior de Ingeniería Agronómica, Universidad Politécnica de Cartagena, 30203 Cartagena, Spain; jorgel.sanchez@upct.es (J.L.S.-N.); erasmo.lopez@upct.es (E.I.L.-B.);

* Correspondence: hayet.kerras@edu.upct.es (H.K.); md.miguel@upct.es (M.D.d.-M.G.); Tel.: +34-968-325-784 (M.D.d.-M.G.)

**Abstract:** Today, the relationship between gender and information and communications technologies (ICTs) is a very important element in achieving sustainable development, since ICTs play a key role in attaining gender equality and empowering women by allowing access to important information and involving them as actors in social, economic and environmental development. This participation is closely linked to the degree of education, training and employability, and so women bring added value to the technology sector and not only to it, but also to all sectors associated with it, through their contribution to R&D and Innovation. The 17 goals adopted in Agenda 21 constitute a roadmap that aims to involve all actors and impose gender equality in each one of these goals. In this study, we compare the innovation and gender index of four Mediterranean countries (France, Spain, Morocco, and Algeria) and analyze how some indexes related with "the gender digital divide" affect the achievement of these sustainable development goals. It has been observed that Sustainable Development Goals (SDG) 2, 3, 4, 5, 8, 9 and 10 are the most influenced by ICT and the gender digital divide, and that none of the countries in our study have achieved them, although France and Spain present a moderate trend towards their achievement by 2030, and to support this statement, a multiple linear regression has been performed at a global level for the countries that have all of the indicators' data available. The empirical results show that the gender digital divide has a negative effect on this accomplishment and that the technology disposition has a positive effect on them.

**Keywords:** sustainable development goals; gender digital divide; equality; information and communication technologies

## 1. Introduction

The technological revolution which emerged at the end of the last century was not just the change from an industrial to a technological model; rather, it was accompanied by a radical transformation in several key fields, namely the economic, labor, geographical and cultural areas.

It is clear that it was the constant technological changes and frequent innovations in information and communications technologies (ICTs) which first enabled the development of the information society. As pointed out by López [1], innovation, viewed as a social system, led to a whole series of changes that spurred on the technological developments now viewed as veritable digital revolutions, those that lay the foundations for the information society. In effect, the development of digital technologies improved the flow of information through the digitized communication of text, sound, voice, images and other multimedia elements.

This exchange of digital information is, in essence, the technological foundation of information societies. In addition, access to ICTs allows citizens to access a greater volume of information and, consequently, to make better and more appropriate decisions in less time, thus providing a competitive advantage.

The United Nations Organization [2] underlines how the lack of access and inappropriate use of technologies slows down the attainment of sustainable development. As an example, it mentions the impossibility of having data analysis available that would help in responding to the occurrence of fatal diseases, consequently leading to a delay in taking important healthcare decisions. It also cites, by way of example, how artificial intelligence allows digital scanners to be read, gaining time and saving lives.

Moreover, the technological revolution is shaking up the business models of enterprises, helping to change uses on an unprecedented scale, transforming the ways goods and services are produced, distributed and consumed.

All these benefits of technological innovation and development currently underscore the need to make efforts to ensure that the use and enjoyment of technology can reach all sectors of the population, avoiding the creation of social inequalities in terms of access, use and appropriation of ICTs described as "Digital Divide."

Hamburg and Lütgen [3] define this general concept of digital divide as "the gap of unequal accessibility and experience for access and use of modern information and communication technologies."

One of the most relevant social inequalities in terms of technology is the lack of women in this sector, something which affects employment, university research, the development of new technologies and, indeed, sustainable development.

Castaño [4] defines "the gender digital divide" as inequalities between men and women in the intensity of use of computer and internet connection as well as in the participation in the basic uses of internet and indicates that this kind of segregation could be measured through equality and ICT indicators.

According to Fountain [5] "The participation of women in the design of information technology at all stages will both straighten up a significant shortfall in human capital and influence the construction of an increasingly information-based society." In this regard, UN [6] points out that women and men are actors in and beneficiaries of development, and women's participation in this is not merely a question of justice and human rights but also a question of economic calculation, because ignoring half of humanity would not make it possible to achieve the desired sustainable development.

The importance given to the use of technology by all countries and all their inhabitants in the same way, pushes us to ask ourselves some questions: How does the inclusion of women in the ICT sector affect sustainable development? What impact do ICTs have on achieving the sustainable development goals? Additionally, which sustainable development objective is most affected by these two factors?

Therefore, this article has focused on the importance of women's participation, as a key element in the implementation of Agenda 21 for Sustainable Development, and analyzes the relationship between the existence of a digital gender gap with the achievement of Sustainable Development Goals. For this reason, a comparison of the achievement of these goals has been made between four countries: France, Spain, Morocco and Algeria, and to support this statement, a multi-linear regression has been performed at a global level for the countries that have all of the indicators' data available.

The selection of these four Mediterranean countries is due to the relationship they maintain, which has always been very close. First, because of their past and shared responsibilities, and aspects such as their transmitted and inherited culture and heritage; and also because of geographical proximity; but above all, because of the economic and trade relationships they maintain.

## 2. Theoretical Framework

Toffler [7] wrote that "we are the last generation of an ancient civilization and the first of a new civilization." Indeed, we are living in a historical time of great technological changes that bring other

changes in the social and economic structure. These transformations, made up of a set of technological innovations that we are experiencing, have been called "the digital revolution." By digital revolution, we refer to technological changes in production and, consequently, in daily life, which have occurred since the 1980s–1990s. The most specific technological changes are related to the massive use of the internet, robotics and digital platforms, the management of Big Data, the application of 3D printing to the industry, and artificial intelligence among others.

These changes have had a significant impact on the lives of humans: residence, mobility, education, political thought, values and attitudes; these have been developed in relation to the changes that occur in the way of producing goods and offering services to companies and individuals. From this, new needs and consumption have arisen that have increased employment, more so than the work positions that disappeared due to the various new technologization waves.

In short, technological convergence is, today, an unquestionable fact. It is introducing an additional saving factor that makes the work of people and of the machines themselves more productive, at the same time allowing an increase in productivity and a reduction in costs. In this sense, Azadnia et al. [8] indicate that: "Topics such as health, careers, business, security, the environment, regional development, human rights, and education are affected by ICT; while in many cases, this connection can be two-way."

However, with the development of the internet and WEB 2.0, the use of ICTs has grown and the vast majority of citizens in industrialized countries use it to access information, creating a digital geographic divide with developing countries, where high-speed internet access is in the reach of most households.

The impact of technology is not the same in all countries, as each has an absorption level at different speeds and with large gaps, both between countries and within themselves. Indeed, the development of ICTs and their appropriation were not done in the same way or at the same speed among all countries. This fracture prevents a large majority from taking advantage of the benefits that globalization has brought [9].

The UN [2] reveals that 80% of the population of developed countries uses technologies, compared to only 50% in developing countries and 20% in the least developed countries. These rates are explained by the resources and capacities (high income) that developed countries have, which are usually the first to implement innovations and, consequently, generate an economic effect that differentiates their profits compared to other countries and creates a "Digital Divide." However, the difference between men and women in having access to ICTs can be seen in many societies, both in so-called economically advanced countries and in those that are developing.

The term "Digital Divide" was first used in the United States in the mid-nineties to refer to the social inequalities that began to emerge as the use of computers and the internet became widespread.

Rogers [10] defines the digital divide as "the gap between individuals who benefit from the internet and those who are relatively disadvantaged with regards to the internet" and relates it to the concept of intellectual capital and the knowledge divide. This concept explains the speed of information appropriation by different populations. Rogers [10] also shows the benefit inherent to innovation and the degree to which this innovation provides benefits.

Along the same lines, Santoyo and Martínez [11], have defined this concept as the separation between people, communities, states, and countries that use information and communication technologies as a routine part of their daily lives and those who do not have access to them, and who, even if they do have them, do not know how to use them.

However, in this regard, Batista [12] links the digital divide in terms of "participation" and "quality of use" and not just the scope of the equipment.

It thus seems reasonable to assert that the existence of a digital divide is directly associated with four elements [13]:

- The availability of a computer (or other hardware element) that allows citizens to connect to the internet.
- The possibility of logging in and accessing the network, from home, work or the office.

- Knowledge of the basic tools for access to "browse" the network.
- Suitable capability to allow the information accessible on the network to be converted into "knowledge" by the user.

Ming-te [14] indicates that the Digital Divide is a very serious matter for those who are currently behind in internet access, for they are not able to enjoy the many benefits of being wired and are handicapped in participating fully in society´s economy. Singh [15] opined in this purpose that: "ICT provides new opportunities for those who are literate, have a good education and adequate resources. Disadvantaged and marginalized groups have little chance to benefit from tools such as the internet automatically. This further increase social divides, widens the gap between rich and poor countries, regions, individuals and even between men and women."

In short, the digital divide is not only measured by use or access to ICTs but also by digital skills and tools. In effect, knowledge and technology are advancing and progressing for everyone, but the problem is that not everyone can access their benefits equally. Among these vulnerable collectives who encounter greater difficulties in accessing technology are women, for whom barriers to access and participation are a notorious reality. This issue has led to undertaking this work, since digital transformation is no longer optional given that it affects all sectors, all institutions, and is in fact detrimental to all those who cannot fully enjoy this transformation.

These kind of digital inequalities are today at the heart of the concerns of several governments and have given rise to several beginnings due to the impact it has on sustainable development. The concept "Gender Digital Divide" refers, according to the Organization for Economic Cooperation and Development (OECD) [16], to the distance between individual, residential, business and geographic areas at different socio-economic levels in relation to their opportunities to access new ICTs, as well as the use of the internet, which ends up reflecting differences both between countries and within them. This concept has been divided into two dimensions or gaps: the first refers to social inequalities that occur as a result of access to ICT infrastructure and tools, while the second refers to characteristics that differentiate citizens who use them or not [17].

In recent years, the presence and participation of women in the field of technology has not improved much, which affects employment, university research, and the development and innovation of new technologies. Therefore, it is necessary to eliminate the multiple labor and social gaps, since it will favor the economic, social and cultural development of our society.

In fact, women and girls are disproportionately affected by a segregation in the access and use of technologies. This "Gender Digital Divide" is due to many factors which handicap their inclusion in the ITC sector.

It is noted today that in many countries, there are fewer women interested in this world of technologies and involved in this sector, and in this, as beneficiaries or protagonists. We also note the existence of a segregation in the choice of IT professions. These stereotypes definitely limit women to some tasks like web page design, database maintenance, and computer security programming, preventing them performing other professions such as those related to hardware or those with more responsibilities like computer designing, because they are considered more characteristic of the masculine gender, due to their strength, or because they require more free time. This masculinization of the profession also impacts the use of these tools, and sometimes makes them more suitable for male use than for female use. Bührer and Schraudner [18] have commented on this purpose that designers assume many technologies to be neutral, but many barriers in use exist, and have given an example of problems related with artefacts of speech recognition systems in artificial intelligence that were not able to recognize female voices, since the designers did not think about the fact that adapting the technology to male voices could exclude female users.

Additionally, there is another factor that influences the quality of use of internet between countries. This factor could be summarized in the lack of training of the predominant language in this field, English, that prevents a good use of these tools. In this sense, Wolk [19] indicates that English

has become the official language of the internet by default and that its handling allows to achieve competitive advantages for developed English speaking countries.

Several alarm signals have been issued by the OECD and the European Union regarding this type of divide, which delays women's joining the world of new technologies, indicating that the exclusion of women in this field represents not only an infringement of a fundamental right but also an infringement of the democratic principle itself.

This is why there is an urgent need to involve not only the public authorities in order to overcome and remove the differences that hinder the fulfilment of fundamental rights, but also specialists to analyze this discrimination as well as representatives of businesses that are not yet sufficiently aware of the gender issue.

In effect, there are various factors that restrict women's confidence in their own abilities to effectively perform in ICT-related sectors, plus other factors that ultimately divert their attention from the possibility of enrolling in these technical careers.

The studies conducted permit removing certain barriers that block the entry of women and their continuity in the technological domain. These factors constitute the cut-off lines of the digital second division. These include human capital, family context, social and/or economic context (employment rate, for example).

In reality, the employment, social and family context plays an important role in driving the vision and ideas of women previously affected by stereotypes relating to feminized and masculinized careers and professions that voluntarily or otherwise create job segregation, and most of the time, this situation is projected onto careers in computer and telecommunications engineering. In this regard, González et al. [20] indicate that "the relationship between women and technologies has historically been reduced to an image of phobia towards technology, which defames women as being against the use of technologies."

Anker [21] distinguishes between two forms of occupational segregation by gender: one Horizontal and one Vertical. The first type of segregation refers to the role assignment imposed by the sexual division of labor, which encourages women to join the healthcare, social, educational, administrative and commercial retail sectors [22]. As for the second type of segregation, it refers to the limitations that prevent women from moving upwards in the corporate hierarchy, which we will define as "The Glass Ceiling."

Segerman-Peck [23] defines the "glass ceiling" as "a set of discriminatory, seemingly invisible mechanisms setting a limit on women's career advancement that is difficult to overcome." In effect, it has been noted over the years that there is a shortage of women in positions of responsibility, unlike men who more easily achieve career promotions. This phenomenon is a reality and a cause for concern that pushes economic specialists to analyze the effects of this occupational imbalance, even when enjoying the same working or job training conditions.

Castaño [24] signals that the information society has enabled the new decisions of women's labor participation to translate into a greater presence in the sector structure, but in terms of access to specific positions and occupations, the traditional segregation between male and female positions still exists. In fact, according to Peng et al. [25] women are expected to take responsibility for domestic chores even when they hold a full-time job. Therefore, they may spend less time and effort on their paid work. Moreover, as they are constrained by household responsibilities and gender stereotypes, employed women are likely to be victims of discrimination in the workplace.

The problem of work-life balance has a major influence on the creation of this professional disparity and makes it more difficult for women to perform and develop leadership tasks, preventing them from holding higher-level positions for which they are as prepared for as their male peers, thus restricting their professional development. The ICT sector is known for long working hours and, above all, for unpredictable schedules and the need for permanent availability.

In some IT positions such as support and maintenance engineers, night-time interventions are needed to ensure the operation of the business during the day (e.g., banks or logistics firms). For some women, especially those with family commitments, this pace is hard to sustain.

The statistics produced by the National Statistics Institute [26], indicated that "women spend 26.5 h per week caring for children or relatives or performing household chores and other unpaid collaborative tasks, compared to 14 h for men."

This lack of availability of women also undermines them when it comes to geographical mobility or supplementary training or attendance at conferences, which, in general, take place outside working hours.

On the other hand, Meykens et al. [27] raised also another type of exclusion, indicating that women often felt excluded from an international career and from further career progression because of gender discrimination.

All of these factors lead to another type of inequality under the title of "gender pay gap," which refers to the difference in gross earnings between men and women. This wage inequality is measured by quantifying the earnings of all women who have worked one year full-time in a country, and of all men in the same situation, with an average being established for each one [28].

According to Becker [29], human capital hypothesis, educational attainment, work experience and on-the-job training are powerful determinants of productivity and, indeed, of individual earnings.

Actually, the pay gap is closely related to horizontal and vertical occupational segregation, and has a direct impact on how women accommodate their work and other personal obligations. In effect, if a woman is not paid the same salary as a man while having more responsibilities than he, she will never strike a work-life balance that will allow her to cover the costs of childcare and/or nursery school, cleaning services, investment in electronic material, etc., so as to be able to devote the same paid working time as men. In that way, Jabbaz et al. [30] indicate in their study that the gender pay gap is greater when woman have children than when they do not have them. In addition, Sánchez-Vidal [31] confirms that the lack of confidence in one's own competences justifies this wage gap given that women think they are less valid than their male counterparts.

Moreover, greater resistance is observed in the use of technologies in some women, which hinders their professional learning and training compared to their male counterparts. This technophobia translates into negative behavior towards machines that always seem difficult to operate, bad, slow or boring; or towards themselves, thinking that they are incapable of using them or that it is very difficult. In this line, Nkosingiphile et al. [32] insist on the importance of training and describe that kind of coaching as a valuable element for women´s development in the Information Technology industry for various reasons, including expanding skillset and on-the-job training; specifically, they mentioned that this element acts in preventing stagnancy either in job position or job performance. It is also observed that there are few female role models in the tech world, with the exception of certain female mentors and networks. Horizontal job segregation is evident in this sector, which is not very favorable to the inclusion of professional women.

According to Botella et al. [33], one of the main reasons is the lack of visibility of women already working in the tech world, discouraging other women to enter the field. This trend decreases the percentage of women, which also reduces their support network and can cause workplace dissatisfaction to arise in the end.

In effect, the female role models in the field of ICTs is a significant factor in the absence of women in this sector. So much so that girls who have no female role models in the tech field show no interest in technological sciences. By way of example, the TGIS [34] study indicates that only 11% of people working in cybersecurity worldwide are women. To counteract the stereotypes of ICTs as a masculine world or as a thing for men, Castaño [35] recommends promoting the presence of female teachers in computer and browser learning courses at all levels, in colleges and universities, in town halls and associations, and especially in courses aimed at women and young people.

In essence, we can summarize the three key elements that positively or negatively affect the gender digital divide in equal access to education, equal access to employment and access to and training in internet use and digital tools; it is the reason why Botella et al. [33] insist in the adoption of gender equality plans which increase the visibility of women in the technology field, promote equal parenting and establish flexible work arrangements, and reduce gender wage gap.

In effect, the contribution of education to lasting development is more than evident, as it enables the development of skills and human capital capable of bringing their country to the desired point of development.

It is clear that countries having more skilled and competent citizens in all areas, and not just scientific and technological ones, is a major contributor to the creation of added economic values. In fact, knowledge clearly contributes to instilling value characterized by openness such as tolerance and equality as well as training in more specific fields, which can enhance the effectiveness of the education system and facilitate job placement.

The Unesco Conference report on education for sustainable development [36] supports this reflection by stating that "Realizing sustainable development requires a widespread change in mindsets and behaviors." This same report also reminds us that "education is essential to promote sustainable development and improve individuals' capabilities to tackle environmental and development problems." With one of the sustainable development goals being guaranteed education, which aims to transmit the necessary techniques, knowhow, values and knowledge, this goal establishes that education should be accessible to all and in all social contexts.

The lack of women in the ICT sector definitely has a huge impact on their contribution to the economy; technology being one of the pillars of sustainable development. In this sense, Gilhooly [37] argued that if countries want to tackle the Sustainable Development Goals seriously, they must appropriately deploy, integrate, and prioritize ICT. The author noted that ICT is essential both to eradicating poverty and in creating sustainable human development.

## 2.1. Influence of ICT on Sustainable Development

Sustainable development is an increasingly popular concept, and its definition and measurement spark a variety of debates and criticism.

Before establishing the relationship between technological development and sustainable development, it is important to remember the definition of this term: Sustainable development aims to satisfy the needs of current generations without compromising those of future generations, according to the "Brundtland Commission" [6].

Sustainable development seeks to create a model of progress that integrates society, environmental protection and the economy alike, balancing these three aspects:

1. Ensuring social equality: Offer the same opportunities to all human communities today and in the future, and enable an improvement in their quality of life (access to employment, education, health, social services, quality accommodation, all kinds of human freedoms as well as integration in society).
2. Maintaining the integrity of the environment: Ensure the protection of the environment in all social, cultural and economic actions so as to maintain the vitality, diversity and reproduction of terrestrial and marine species and natural ecosystems.
3. Improving economic efficiency: Namely, favor the optimal management of human, natural and financial resources in order to meet the needs of human communities. This goal is achieved through the joint responsibility of businesses and consumers, in terms of the production of goods and services and the adoption of appropriate governmental policies.

This concept of sustainable development has been embodied in the program called "Agenda 21" which are planning actions for the 21st century and approved at the Rio Summit in 1992. Subsequently, the OECD [38] supported the relationship of sustainable development with these three dimensions

and indicated that the concept of sustainability refers to the management of natural resources in a way that respects reproductive capabilities. In the social sciences field, this concept goes beyond growth, addressing economic and material wellbeing to encompass considerations of fairness and social cohesion.

Innovation and sustainability have never been seen as complementary concepts, but innovation has always been recognized as a key factor in economic and social development and an essential tool for growth and job creation. Torre-Marin [39] confirms this theory indicating that technological innovation is a method which allows to promote efficiency in energy use, to increase the intensity of materials and to look for coefficients.

This lack of complementarity has been due to the perception of ICTs and other innovations, which has acted as a barrier to the preservation of the environment's integrity. In fact, several authors have taken an interest in the relationship between ICTs and sustainable development [40–50].

Breuil et al. [42] indicate that ICTs are tools for sustainable development and are a resource common to the entire economy. This first theory considers ICTs as an information support element and a communication tool.

Conversely, other authors [46,51,52] view ICTs as a product that has an impact and consequences on sustainable development, since a close relationship between them is reflected in all sustainable development goals.

Forsooth, continued development, observed in recent years, is closely related to technological advances, new inventions and the thrust in R&D and Innovation.

In this regard, Esser [53] indicates that having a good technological policy is one of the critical factors in competitiveness.

Porter [54], meanwhile, emphasizes the importance of technological development in achieving a competitive advantage, through improving the product and the production process. Generally speaking, the economy grows when its economic players (businesses and institutions) have ever more capital. For businesses to generate this capital, their workers need to perform better and more efficiently in the workplace. The key to this growth will thus be investment by businesses in training and in technology resources.

ICTs have led to investment growth and, in fact, to an increase in the workforce, with the creation of new jobs, an increase in productivity and improvements in the social situation, with the development of the healthcare, education and security sectors that increase citizens' wellbeing. According to Marlon-Raudales [55], ITC can transform our world. If it is used properly, IT can create economic opportunities, foster social and political inclusion and contribute to shared prosperity.

In reality, countries that do not incorporate technological innovations into their sectors of activity (agriculture, environment, education, trade, transport, etc.) ultimately lose market share and consequently fail to be competitive (productivity, efficiency, innovation, learning, industrial revolution, market expansion, social welfare, new sources of employment, etc.). This theory was confirmed by Ali et al. [56], who says that advanced manufacturing technology facilitates high production with fewer employees and can be readily adopted by competing firms, maintaining standard levels of employee productivity within the industry.

It is therefore important to integrate new technologies into the labor and production process and to start up an education and training plan that allows economic actors to provide added value and fully enjoy this technological progress, which entails improvements in the way goods and services are produced and marketed.

Recently, the CIRIEC General Assembly [57] stated:

> Digital technologies and artificial intelligence offer a great opportunity to face the challenges of the future by creating new relationships between organizations and between people, through, for example, the implementation of shared projects, the improvement of citizen services, the development of the collaborative economy, socially responsible finances, the organization of new solidarity actions, the implementation of new forms of democratic governance,

the exercise of citizenship, etc. They also offer people the opportunity to be actors of their future, in the form of collective public companies, social and cooperative economy, through the appropriation of economic mechanisms and institutions, instead of being only passive consumers, as encouraged by the global giants of international trade.

As explained earlier, ICTs are the key to sustainable development, and this can be observed in the link between them and each one of the 17 objectives and goals [58] in Table 1.

**Table 1.** Link between ICTs and SDGs.

| SDGs | Definition of the SDGs | Link to ICTs |
|---|---|---|
| SDG 1 | End poverty in all its forms everywhere. | Digitization and the development of various computerized financial services clearly contribute to the fight against corruption on the one hand, and against poverty on the other. In this sense, Henry [59] indicates: "More countries and enterprises are recognizing the importance of ICT to poverty alleviation. A poor woman farmer in a rural community can use a telephone to enquire about her vegetable prices without the overhead of a bus fare or the physical endurance such journey requires. One will agree, however; that such possibilities arise out of the willingness to invest financial resources, ICT infrastructure, and capacity building." |
| SDG 2 | End hunger, achieve food security and improve nutrition and promote agriculture. | The incorporation of technological innovation in the domain of agriculture serves to increase yield and feed the entire population (better access to information and, in fact, better decision-making, daily and hourly weather forecasts, more developed tools to promote productivity and efficiency). ICTs contribute to the improvement of food security and the promotion of agricultural sustainability by offering opportunities that benefit farmers, connecting them with remote areas and helping them to improve their farming methods and productivity (better production, market price information, environmental conditions control, food supplies monitoring, delivery efficiency). In this way, Awuor et al. [60] indicate that smart farming for agriculture makes a tremendous contribution to food sustainability. |
| SDG 3 | Ensure healthy lives and promote wellbeing for all at all ages. | ICTs play an important role in the development of the medical industry. In some hospitals, healthcare research centers or medical centers, the information is computerized (Big Data) and the tools developed make it possible to more effectively identify patients' diseases and, consequently, propose more suitable medication. As an example, we can see the importance that is playing in technology (research, awareness and fight) in the case of the appearance of an unknown disease at the present time: the coronavirus. Thanks to technological progress, all countries using ICTs are at the same level of information, including all inhabitants, such as ordinary citizens, medical researchers or governors. Besides, technology encourages active and healthy lifestyles and allows to monitor physical activities to improve the health of those who need exercises such as obese people, and also facilitates clinical intervention [61]. |
| SDG 4 | Ensure inclusive and equitable quality education and promote lifelong learning opportunities for all. | In terms of education, access to ICTs is, on the one hand, allowing for online training, thereby saving time and money, and on the other, for the use of tools and devices that are more powerful and better adapted to the learning needs of the current era. It should be noted that this use and access to technology contributes to the fight against various types of exclusions and segregations. Campos and Navarro [62] indicate that the use of ICTs is being a decisive challenge in the world of education, the advances they offer are ideal to facilitate the participation of the different groups of people involved in the educational process. |

**Table 1.** *Cont.*

| SDGs | Definition of the SDGs | Link to ICTs |
|---|---|---|
| SDG 5 | Achieve gender equality and empower all women and girls. | The use and access to ICTs makes a very clear contribution to the fight against gender inequalities. This is because it allows "its users" access to information and online learning, training and work as an alternative to the issue of work-life balance. Technology also offers women a range of opportunities for empowerment. In this purpose, UN [63] indicates that countries which seek to direct their science, technology and innovation policies towards sustainable development should also consider giving social problems a central place in their policies. Innovation policies that take into account gender issues can favor the participation of women as innovators or entrepreneurs, and youth-centered policies can also contribute to making technological change more inclusive. However, it is important to note that several inequalities, among them online violence directed at women, constitute an obstacle to the participation of women in the public life and have alarming concrete consequences. In fact, the harassment, intimidation and insults on social networks they experience have profound repercussions in the daily life of women and girls [64]. |
| SDG 6 | Ensure availability and sustainable management of water and sanitation for all. | The incorporation of technological innovation in water sanitation plays a very important role and allows for effective quality management of the supply and an equitable and sustainable extension of water. Moe and Rheingans [65] confirm the role of ICTs in achieving this objective and indicate that to face the challenge of water scarcity, it is important to manage three main areas: The first one is improving the efficiency of agricultural, industrial and domestic water use; the second one consists of developing technology to implement and monitor safe water reuse; and the third one relies on the development of technologies and economic policies to promote effective water conservation. |
| SDG 7 | Ensure access to affordable, reliable, sustainable and modern energy for all. | ICTs play an essential role in creating environmental balance and provide energy efficiency, firstly because the role of information in raising awareness is important, and secondly because more and more technological solutions are being put forward every day, such as industrial networks, smart buildings, innovative logistics modes, etc. Expanding infrastructure and upgrading technology to provide clean energy can help the environment by reducing greenhouse gas (GHG) emissions, environmental degradation, and natural resource depletion [66]. |
| SDG 8 | Foster sustained, inclusive and sustainable economic growth, full and productive employment and decent work for all. | The development of innovation-based strategies and ICTs has been established as evidence to promote economic growth and create new jobs, which imposes the necessity for computer knowledge and skills on most jobs. Technology has also brought about many transformations in the way we do business (e-commerce, e-banking, digital marketing, etc.). Sastre-Centeno and Inglada-Galiana [67] indicate that the population can communicate instantaneously and at a limited cost, and this is thanks to the universalization of the internet, which has empowered the interconnection of different networks creating a collaborative economy. On the other hand, Maluck et al. [68] confirm that information technologies (IT) allow the creation of conditions for better position of companies in international markets. |
| SDG 9 | Build resilient infrastructure, promote inclusive and sustainable industrialization and foster innovation. | The development of technological infrastructure and training in the use of the internet are two essential elements to strengthen industry and facilitate the achievement of sustainable development. In that regard, the UN [69] says: "Investments in infrastructure—transport, irrigation, energy and information and communication technology—are crucial to achieving sustainable development and empowering communities in many countries. It has long been recognized that growth in productivity and incomes, and improvements in health and education outcomes require investment in infrastructure." In fact, it has become essential for countries who want to promote sustainable economic development to invest in the improvement of the connectivity in the most isolated areas, and in ICT training for all the population, to obtain the performance desired [70]. |

**Table 1.** *Cont.*

| SDGs | Definition of the SDGs | Link to ICTs |
|---|---|---|
| **SDG 10** | Reduce inequality within and among countries. | Despite the different digital divides in existence, ICTs can be a key element in reducing inequalities between regions and countries. Kolk [71] indicates that: "There has developed a vast interwoven network of reciprocal interests, of open communications for exchange of technology and commercial intelligence, of personnel and cultural patterns." |
| **SDG 11** | Make cities and human settlements inclusive, safe, resilient and sustainable. | The eleventh sustainable development goal is to make cities and human settlements inclusive, safe, resilient and sustainable, and this is effectively what ICTs can achieve through the innovative and intelligent solutions they propound, such as water management, logistical advances, the development of energy consumption management and waste management. According to the United Nations [72]: "The ICT have triggered comprehensive and profound changes for urban development and human settlements, which is reshaping our cities and communities." Sousa et al. [73] also argued that digital technologies allow to build smart cities and make more efficient some areas, such as energy, environment, governance, mobility, buildings and services to improve the quality of life of the citizens. |
| **SDG 12** | Ensure sustainable consumption and production patterns. | With the contribution made by technologies, it is now easy to achieve sustainable consumption and production thanks to the different technical solutions that regulate responsible and sustainable production and consumption (smart grids, cloud computing, etc.). To this end, certain authors such as Breuil [42] argue that ICTs make industries consume fewer resources and encourage dematerialization. Garzón-Castrillon [74] also indicates that, first, it is important to highlight that companies can generate value by reducing the level of material consumption and pollution associated with rapid industrialization. Second, companies can create value by operating at higher levels of transparency and responsiveness, driven by civil society. Third, they can do so by developing new technologies that reduce the size of the human footprint on the planet. |
| **SDG 13** | Take urgent action to combat climate change and its impacts. | The role of ICTs is central in ensuring climate balance and protection, and it is thanks to developed rainwater recovery systems, for example, to weather forecasts, predictions and alerts, or to human awareness of the issue. In this sense, Okokondem [75] insists on the role of space technologies and indicates that "they have led to several inventions that benefit the environment and save energy. Vehicles' carbon dioxide emissions are being reduced by satellite-based systems, wind turbines are more efficient as a result of remote sensing technology and solar cells produce more energy based on information from weather satellites." |
| **SDG 14** | Conserve and sustainably use the oceans, seas and marine resources for sustainable development. | ICTs also play a role in the conservation and sustainable use of the oceans. Satellite monitoring makes it easier to obtain accurate information and data (pollution, meteorological impact, ecosystem status, etc.), which allows for their conservation. IUCN [76] underlines the importance of the use of technologies in activities related to the oceans, either in terms of maritime transport, or in the terms of creating opportunities for greater access and a healthier use of its resources. |
| **SDG 15** | Protect, restore and promote sustainable use of terrestrial ecosystems, sustainably manage forests, combat desertification, halt and reverse land degradation and halt biodiversity loss. | The efficient use of technological innovation also contributes to the conservation of terrestrial ecosystems and the protection of biodiversity. As in SDG14, the role of satellite information and reporting provides scope for anticipating and acting, leading to a more sustainable use of satellites. Petit [44], Faucheux and Nicolas [77] and Breuil [42] support this point by indicating that ICTs play a role in replacing transport and optimizing logistics, and, in fact, have a positive impact on the ecosystem. According to Battol et al. [78] and Andreopoulou [79], ICTs have a key role in the environmental sustainability, since they contribute to reducing carbon emissions and on the protection and the promotion of rural sustainable development. |

**Table 1.** *Cont.*

| SDGs | Definition of the SDGs | Link to ICTs |
|---|---|---|
| **SDG 16** | Promote peaceful and inclusive societies for sustainable development, provide access to justice for all and build effective, accountable and inclusive institutions at all levels. | The involvement of ICTs in the access to justice and the spread of peace is also evident. Access to and communication of information means that everyone can participate in a transparent and equal way in political, demographic, social and cultural processes. Examples include natural disaster situations in which real-time information transmission makes aid and emergency management more effective. Thanks to the technology progress, the electronic justice emerges as a new tool that provides people more convenient access to justice than the conventional manner. Yikun et al. [80] have witnessed the establishment of an electronic-government that offers society great opportunities to receive information about justice services, and indicates that electronic justice emerges as a necessary sector of the electronic administration, especially since it contributes a lot to the constitution of a modern justice administration system and the openness in public services. On the other hand, ICT can lead to increased democratization and enable citizens to participate in the decision-making process of the government, and can also be a good tool for increasing transparency in the country [81]. |
| **SDG 17** | Strengthen means of implementation and revitalize the global partnership for sustainable development. | ICTs support the scope of sustainable development through the creation of relationships and, further, of common goals for humanity, which are summarized in economic growth, social inclusion and environmental sustainability. Development co-operation modalities can include financial transfers, capacity building, technology development and transfer on voluntary and mutually-agreed terms, policy change (for example, to ensure coherence of domestic policies and help to address global systemic issues), and multi-stakeholder partnerships [82]. |

**Source**: Produced by the author.

However, there are other studies [43,46,51] that contradict this analysis and indicate that this theory is nothing more than a myth of an information society. Furthermore, as indicated earlier, the development of ICTs is not only a tool to achieve the 17 sustainable development goals but also a consequence of this development. In effect, implementing each one of these goals reduces inequalities and exclusions in this field.

*2.2. Sustainable Development Goals Selection*

This work focuses on seven of these Sustainable Development Goals (SDGs), which, at first sight, are supposed to be more related and linked to the "gender digital divide," namely: SDG2, SDG3, SDG4, SDG5, SDG8, SDG9 and SDG10. This selection will be verified and commented on in the analytical part of our investigation.

The choice of SDG2 is mainly due to the role of rural women in ensuring food security and in the need to empower them in technological terms to achieve this objective. In this sense, at the World Summit [83], it was indicated that there were several aspects that affected women who lived in rural areas, and the importance of ensuring equal access to goods and resources, like technology, was cited.

The case of rural women in agriculture is not the only one affected by this aspect, the role of women in the health of rural families could be cited in the same way, and it is the reason the SDG3 was selected. Indeed, the most prepared women and those with the most developed digital capacities are abler to better break the glass ceiling imposed on them and reach health posts with high responsibilities that allow them to contribute something in this world.

The fourth goal (SDG4) is complementary to the fifth (SDG5) since, in its targets, and especially goal SDG 4.3, it aims to ensure equal access for all men and women to quality technical, vocational and higher education, including university education.

This goal is related to all the others and makes it possible to increase the skills of human capital in several sectors. This relationship is observed, for example, in the achievement of objectives 2 and 3,

and if it is achieved in parity, it allows even more to reach full employment and full production in the sanitary and agrarian sectors.

In addition, SDG5 pays particular attention to the fight against violence and the generalization of gender equality and the empowerment of women and girls. This means not only offering them equal learning opportunities but also employability and capability-building opportunities that will allow them to access high-level positions, or encourage entrepreneurship to create their own enterprises, on an equal footing with men, and indeed, fight against wage inequalities, which enables economic growth, as SDG8 aims to achieve. In this sense, Gupta et al. [84] affirm that women's participation in entrepreneurship is considerably lower than men in almost all societies, and explain that this is because entrepreneurship is viewed as a male-typed career.

As regards to SDG9, it is closely related to the third point. In effect, it is through investment in technological infrastructures and innovative material that access to and use of the internet and digital tools can become widespread worldwide, bringing added value to the economy and enabling development.

According to an OECD [38] report:

There is broad consensus that the disconnect between economic growth and social wellbeing is increasing. At the same time, research and innovation have become some of the main drivers of growth. However, these two trends could not be reconciled: we are clearly not making the most of innovative solutions aimed at addressing social problems, and society pays a high cost for this in lost opportunities. Social innovation offers a way to reconcile these two forces, generating economic growth and social value at the same time.

Villatoro [85] confirms this theory, indicating that a number of countries have sought to realize the promise of development and wellbeing associated with more widespread introduction of ICTs by implementing nation-wide initiatives intended for universal internet access.

As regards to SDG10, it addresses the reduction of inequalities in a general way and, in fact, relates specifically to the reduction of gaps, whether of gender or between geographical areas.

In this regard, Piketty [86] indicates that 21st-century economic inequality is growing at a rapid and dangerous pace, leading to poverty and social exclusion in several countries.

To conclude, we take the summary by Samuelson [87] as a reference, which confirms the complementarity existing between all these points, indicating that five key elements affect development in general terms:

1.  Human resources: This point relates to job availability and human capital, with all the elements this entails: recruitment, education, training, motivation, remuneration, etc.
2.  Natural resources: Natural resources refers to all the components of the environment.
3.  Capital formation: Namely, all the machinery, factories and roads that contribute to the creation of capital.
4.  Science and technology: This includes all efforts made in this field, such as R&D and Innovation in engineering, for example.
5.  Legal stability: This is about the effectiveness and application of laws that ensure stability.

The complementarity between these elements has been confirmed by Wang et al. [88] who resumed the ways to deconstruct gendered technology by allowing more women to enter science and technology, by making good use of women´s characteristics, and finally, by developing a female alternative science and technology.

## 3. Comparative Analysis between the European Union and the Maghreb

This analysis aims to compare and analyze the impact of the gender digital divide on sustainable development in the European Union (EU) and the Maghreb, taking two countries from each region as a reference: France and Spain to represent the EU, Morocco and Algeria to represent the Maghreb.

The selection of these countries is due not only to the geographical and historical proximity that unites them, but also because of the commercial and economic relations that they maintain. The two countries of the European Union have had a presence in both Maghreb countries for centuries and have left cultural and social traces that make Morocco and Algeria quite linked to them. This impact is noted by the "Darija" language used, very marked by both languages. This social impact is also reflected in the way of being and living (gastronomy, festivals, traditions, habits, etc.) that is closer to the European than that of the Middle East.

This closeness is also noted by the number of immigrants in the two countries, as well as by the number of preceding French and Spanish companies in each of the two countries. Indeed, all four countries are very strategic partners and are turning the Mediterranean into an important economic hub. In effect, the Maghreb region, particularly Morocco and Algeria, are at the heart of European foreign trade priorities. These relationships have mainly been based on the supply of energy, gas and oil.

In this sense, the Euro-Mediterranean Network of Economic Studies [89] is committed to promoting trade and investment among neighboring countries, in order to contribute to economic growth in the region. As a matter of fact, it aims to strengthen exchanges between the European Union and its southern periphery through the signing of free trade and reciprocal exchange agreements that stimulate competition, promote investment and ensure technology transfer [90].

As well as this, the Union for the Mediterranean [91] highlights the importance of strengthening Mediterranean relations and promoting exchanges of information and good practices between the two shores, and comments that this contributes to promote the technology transfers in order to create employment opportunities, and seek the participation and contribution of women and youth in the development of production and consumption-oriented models aimed towards achieving sustainable development.

However, despite their vicinity, shared history and common interests, these countries have not developed at the same pace, owing to a number of factors and contexts that emerged during their respective histories (backwardness due to wars, need to rebuild all infrastructures and to educate an entire people, period of terrorism, etc.), and which have forced each country to focus on different economic, social and environmental priorities.

In fact, achieving sustainable development is the global challenge that aims to tackle global problems and improve human wellbeing. However, the latest studies carried out to measure the progress made towards their goals reveal the enormous difficulties that countries face in achieving them.

The Sustainable Development Report [92] indicates that, so far, no country in the world has achieved the 17 sustainable development goals and assumes that none will have attained them by 2030, although it does show in its ranking that some countries have made more progress than others. At the top of the ranking, developed countries such as Sweden (84.8%), Denmark (84.7%) and Finland (82.2%) stand out. In contrast, it denotes that the Central African Republic, Chad and the Democratic Republic of Congo are at the bottom of these rankings.

With regards to the four Mediterranean countries covered by this research, the following classification is highlighted (Table 2).

**Table 2.** Ranking of sustainable development in France, Spain, Morocco and Algeria.

| Country | Ranking | Punctuation |
|---------|---------|-------------|
| **France** | 4 | 81.5% |
| **Spain** | 21 | 77.5% |
| **Morocco** | 72 | 69% |
| **Algeria** | 53 | 71.1% |

**Source:** Own elaboration from Sustainable Development Report [92].

Table 2 shows that France ranks in 4th place out of 162 countries committed to implementing sustainable development goals. The report by Bertelsmann Stiftung [92] indicates that the country scored 81.5%. As for Spain, even though it lags behind other European countries, it ranks 21st with a score of 77.5%. Morocco and Algeria, however, are more backward, occupying positions 72 and 53 respectively, with 69% and 71.1%.

With regards to the score for each one of the goals, we can see the differences between the countries in Table 3.

**Table 3.** Scoreboard for sustainable development goals of France, Spain, Morocco and Algeria.

| Country | France | Spain | Morocco | Algeria |
|---------|--------|-------|---------|---------|
| **SDG 1** | 99.5 | 98.1 | 94.9 | 97.8 |
| **SDG 2** | 66.0 | 56.2 | 53.8 | 52.7 |
| **SDG 3** | 94.3 | 95.4 | 73.7 | 75.5 |
| **SDG 4** | 97.4 | 95.4 | 78.0 | 85.9 |
| **SDG 5** | 86.5 | 82.7 | 42.9 | 51.1 |
| **SDG 6** | 87.9 | 88.1 | 66.1 | 63.6 |
| **SDG 7** | 97.0 | 94.7 | 87.7 | 85.9 |
| **SDG 8** | 78.1 | 75.2 | 67.4 | 69.7 |
| **SDG 9** | 73.6 | 68.1 | 32.4 | 29.8 |
| **SDG 10** | 85.6 | 69.2 | 61.5 | 88.7 |
| **SDG 11** | 87.0 | 89.1 | 72.2 | 66.6 |
| **SDG 12** | 53.4 | 53.4 | 82.5 | 86.5 |
| **SDG 13** | 86.4 | 93.3 | 92.4 | 94.3 |
| **SDG 14** | 64.2 | 59.4 | 48.2 | 41.9 |
| **SDG 15** | 76.7 | 65.4 | 75.6 | 63.2 |
| **SDG 16** | 76.6 | 80.6 | 69.0 | 72.4 |
| **SDG 17** | 75.1 | 59.1 | 75.9 | 83.0 |

**Source:** Own elaboration from Sustainable Development Report [92].

The report by Bertelsmann Stiftung [92] tells us that France obtains its best result in SDG 1, but also very good results in SDGs 3, 4 and 7. However, France generates significant negative environmental and security externalities, which is due to its trade activities, including arms exports, which jeopardize the ability of other countries to attain the SDGs. This same report indicates that **Spain** has its best result in SDG 1 (98.1%) and very good results in SDGs 3, 4, 7 and 13. However, this same report indicates that more efforts need to be made to ensure equal opportunities for the whole population and to attain the other goals. It has made the least progress in SDGs 2, 12, 14 and 17. Like France, Spain, too, generates negative environmental externalities that undermine the capacity of other countries (mainly in Latin America) to attain the SDGs. For its part, Morocco forms part of Africa's leading countries in terms of sustainable development and demonstrates the results of various efforts made towards attaining several objectives in SDGs 7 and 12, with an excellent result highlighted in SDGs 1 and 13. However, there is still some leeway for improving the situation of SDGs 9 and 14. As for Algeria, the report indicates that it occupies top position in the ranking of Arab League countries and African countries, although it has a long way to go before it catches up with the most developed countries. Some very important advances are highlighted in SDGs 1 and 13 and some other good results in SDGs 4, 7, 10, 12, but Algeria faces many difficulties in improving its results in SDGs 9 and 14.

Below, we can see the progress made by each country in terms of attaining the sustainable development goals (Figure 1).

| SDG | France | | Spain | | Morocco | | Algeria | |
|---|---|---|---|---|---|---|---|---|
| | Situation | Evolution | Situation | Evolution | Situation | Evolution | Situation | Evolution |
| SDG 1 | ⬤ | ⬆ | ○ | ⬈ | ○ | ⬆ | ○ | ⬆ |
| SDG 2 | ○ | ⬈ | ○ | ⬈ | ○ | ⬈ | ○ | ⮕ |
| SDG 3 | ○ | ⬆ | ○ | ⬆ | ○ | ⬈ | ○ | ⬈ |
| SDG 4 | ○ | ⬆ | ○ | ⬆ | ○ | ⬈ | ○ | ⮕ |
| SDG 5 | ○ | ⬈ | ○ | ⬈ | ○ | ⬈ | ○ | ⬈ |
| SDG 6 | ○ | ⬈ | ○ | ⬆ | ○ | ⬈ | ○ | ⬈ |
| SDG 7 | ○ | ⬈ | ○ | ⬆ | ○ | ⬈ | ○ | ⬈ |
| SDG 8 | ○ | ⬈ | ○ | ⬆ | ○ | ⬈ | ○ | ⮕ |
| SDG 9 | ○ | ⬆ | ○ | ⬈ | ○ | ⬈ | ○ | ⬆ |
| SDG 10 | ○ | ⬆ | ○ | ⮕ | ○ | ⬤ | ○ | ⬤ |
| SDG 11 | ○ | ⬈ | ○ | ⬈ | ○ | ⮕ | ○ | ⮕ |
| SDG 12 | ○ | ⬤ | ○ | ⬤ | ○ | ⬤ | ○ | ⬤ |
| SDG 13 | ○ | ⮕ | ○ | ⮕ | ○ | ⬆ | ○ | ⮕ |
| SDG 14 | ○ | ⬈ | ○ | ⬈ | ○ | ⮕ | ○ | ⮕ |
| SDG 15 | ○ | ⬈ | ○ | ⮕ | ○ | ⮕ | ○ | ⮕ |
| SDG 16 | ○ | ⬈ | ○ | ⬈ | ○ | ⬈ | ○ | ⮕ |
| SDG 17 | ○ | ⮕ | ○ | ⮕ | ○ | ⬤ | ○ | ⬤ |

○ SDG achievement  ○ Challenges remains  ○ Significant challenges remain  ○ Major challenges remain

⬆ On track  ⬈ Moderately increasing  ⮕ Stagnating  ⬇ Decreasing  ⬤ Data not available

**Figure 1.** Comparison of the scope of the sustainable development goals (2019). (**Source:** Own elaboration from Sustainable Development Report [92]).

Moreover, and as indicated earlier, this study is interested in comparing between 7 of the 17 goals (SDGs 2, 3, 4, 5, 8, 9, 10) associated with the gender digital divide, and so we summarize this situation in Figure 2.

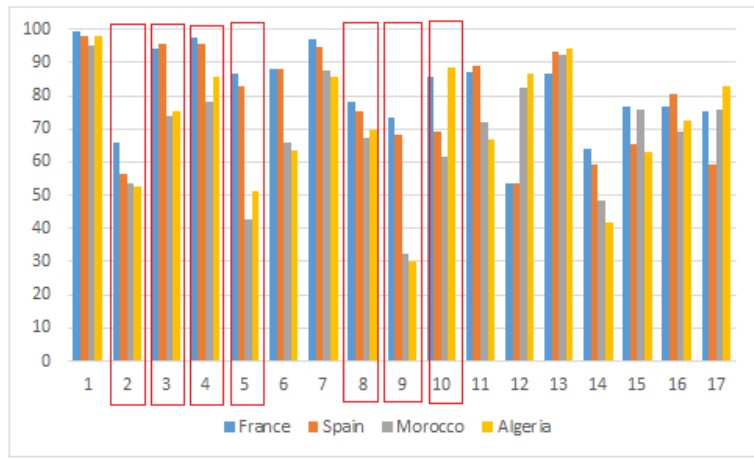

**Figure 2.** Comparison between the progress of each country in terms of relevant SDGs: SDG 2, 3, 4, 5, 8, 9, 10 (2019). (**Source:** Own elaboration).

In Figure 2, we see that in terms of ending hunger (SDG2), the four countries are far from reaching the objective. Nevertheless, France predominates with 66%, Spain with 56.2%, followed by Morocco and Algeria which reach 53.8% and 52.7%, respectively. This situation is due to the difficulties and less developed agricultural infrastructure that the two Maghreb countries have.

With regard to the objective of ensuring good health and wellbeing for all, it is quickly observed that Spain reigns in this area with a result of 95.4%, followed closely by France, which has a score of 94.3%, then Algeria with 75.5% and Morocco with 73.7%. It goes without saying that the socio-political problems marked by the two Maghreb countries have delayed them a lot, and also mean that they do not have the same tools to compete with the countries of the European Union.

As to quality of education (SDG4), the two European countries (France, 97.4%, Spain, 95.4%) are more developed than the two Maghrebi countries (Morocco, 78%, Algeria, 85.9%). This situation is explained, on one hand, by the historical context that made it impossible to uphold educational standards in the two Maghrebi countries (colonization, the need to rebuild infrastructures and to rebuild the two countries after wars) and, on the other, by the development of resources, including technology, which allows the two European countries to make greater advances.

In the case of gender equality (SDG5), the same effect is observed, but with lower results for the four countries (France 86.5%, Spain 82.7%, Morocco 42.9%, Algeria 51.1%). In effect, despite the efforts made around the world, and more so in European Union countries, the issue of gender equality and the empowerment of girls and women has not yet become evident in societies. In the Maghreb, this issue has had a significant impact due to the extent of social development and other cultural factors (among them the interpretation of religion), which is not part of the government's priorities, although some changes have actually been made thanks to awareness-raising by many NGOs.

This theory was confirmed by some authors [93,94] who indicate that the reason for differences in business ethics across countries may well relate to differences in cultures, since culture affects moral orientations such as idealism and relativism.

With regards to SDG8, which refers to the promotion of sustained, inclusive and sustainable economic growth, the graph shows greater development in the case of the two European countries compared to their Maghrebi neighbours (Spain 78.1%, France 75.2%, Morocco 67.4% and Algeria 69.7%). Nevertheless, we can also see that the disparities are not as great as in the previous two SDGs. This is due to the economic plans drawn up in each country to secure strategies that allow for job creation, and to investments made in the field of technology to bring about transformations in the way they do business.

Furthermore, the graph shows that France again overtakes the three other countries in terms of industry, innovation and infrastructure (SDG 9) with 73.6%, followed by Spain with 68.1%, and then Morocco with 32.4%, and finally, Algeria with 29.8%. This second-last social development goal is evident in Morocco, with more developed technological infrastructures and more substantial investments in industry; plus, the strategy of diversification of resources it has developed and its openness to the world and diversification of foreign partners. This contrasts with Algeria, which has always been the victim of its energy wealth and has greater difficulty in developing other sectors, and particularly that of technology. This result also shows that the percentages for the two European countries more than double the other two, indicating the presence of a very large digital divide between countries caused by the ability to access and use the internet but also in educating and training people in this use.

Finally, it is notable that when it comes to inequalities between regions and countries (SDG 10), the ranking order changes. Algeria is ahead with 88.7%, followed by France with 85.6%, then Spain with 69.2% and finally, Morocco with 61.5%. This situation can be explained in the case of Algeria and France by the fact that they have the same general law in all departments and regions, unlike Spain, which is governed by the laws of the autonomous regions, meaning that the same conditions and opportunities are not available in all regions and that some areas are, in fact, penalized. In the case of Morocco, too, the situation can be explained by the existence of two predominant social classes, also by

having this social (and sometimes economic) separation between the northern cities, more impacted or influenced by the Spanish, and southern cities, which are affected by the French.

Next, in Table 4, we set out a comparison in terms of the Human Development Index (HDI), a statistical index that takes into account criteria such as life expectancy, health and education, and of another one, Economic Development (GDP), which is an economic indicator for quantifying the total value of annual wealth production.

**Table 4.** Comparison in terms of HDI and GDP (2019).

| Country | IDH | GDP |
| --- | --- | --- |
| **France** | 8 | 6 |
| **Spain** | 15 | 13 |
| **Morocco** | 118 | 57 |
| **Algeria** | 100 | 52 |

**Source:** Own elaboration from FMI [95] and PNUD [96].

Table 4 analyzes the ranking of Mediterranean reference countries according to a social indicator and another economic indicator in order to see how investment in innovation (infrastructure, training, connectivity, etc.), and how the involvement of women in the economic world, in general, can influence the country's overall performance, whether in terms of social or economic development.

This shows a ranking of countries in terms of HDI first, with France ranking 8 and thus, in the world's top 10 countries in terms of life expectancy, health and education. As for GDP (Gross Domestic Product), France ranks 6th, with its GDP amounting to USD 2,840,000.

We also observe that Spain is among the top 20 countries in terms of HDI and ranks 15th, with GDP in 13th position, equivalent to USD 1,470,000.

As regards to Morocco, it is quite backward in terms of the human development index, ranking 118th, with GDP positioning at 57 with a mere USD 122,460.

Finally, we observe that Algeria occupies position 100 in terms of social index, with a GDP equivalent to USD 200,170, which puts it in position 52. In fact, as several authors have argued, the earlier argued theory that technology is positively associated with economic development is confirmed, and more so when it is used by several categories of persons, regardless of their race, age, educational level or sex.

It is therefore essential to invest in everyone's learning and training, giving people the same possibilities and opportunities of progress and employability without detriment or assignment of roles based on the stereotypes that society has unconsciously imposed.

Meanwhile, as analyzed above, and perfectly reflected in the case of Morocco and Algeria, investment in technology in general (infrastructure, material, training, R & D and Innovation) is the essential key to the economic, social and environmental growth required to achieve the 17 SDGs and meet the set challenge.

To complete this analysis of the impact made by the gender digital divide on sustainable development, it is imperative to also compare two other important elements, which are the innovation indicator that classifies the innovation results of 130 countries according to more than 80 indicators, and the gender gap indicator that measures the magnitude of the gap between women and men in terms of health, education, economy and political indicators (see Table 5).

**Table 5.** Comparison by innovation indicator (2019) and digital divide indicator (2018).

| Country | Innovation Indicator | | Gender Gap Indicator | |
|---|---|---|---|---|
| | Ranking | Punctuation | Ranking | Gap Index |
| France | 16 | 54.25 | 12 | 77.90% |
| Spain | 29 | 47.85 | 29 | 74.60% |
| Morocco | 74 | 31.63 | 137 | 60.70% |
| Algeria | 113 | 23.98 | 128 | 62.90% |

**Source:** Own elaboration from Global Innovation Index [97], Gender gap indicator [98].

Table 5 again confirms the above results for innovation, highlighting that in terms of development in this field, France is one of the leaders in position 16, with Spain ranking 29th. Although it is not one of the top 20 technology pioneers, it is well-positioned among the 162 countries. Banegas and Myro [99] commented, in this sense, that in the last years, Spanish companies have developed significantly above all on the tertiary sector and have made great efforts to reorganize and innovate. As for Algeria and Morocco, they are quite backward, occupying positions 74 and 113, again demonstrating Morocco's involvement in the development of other sectors, which make up its main resources.

Additionally, the report on the Gender Gap Index tells us that France has a gender gap of 77.9%, which puts it among the countries with the fewest differences between men and women. This Index analyzes the division of resources and opportunities between both sexes in 149 countries and measures the size of the gender gap in participation in the economy and in the world of skilled work, in politics, in access to education and in life expectancy.

Spain has a gender gap of 74.6%, putting it in 29th position.

Morocco, by contrast, has a gender gap of 60.7%, positioning it among the countries with the greatest differences between men and women, affecting the country's competitiveness.

Finally, the gender gap percentage in the case of Algeria is 62.9%, which puts it at 128. Algeria, which is 2.2 points ahead of Morocco, is also among the countries with the largest gender gap.

This relationship between technology and gender can be determined in a more specific way, taking into account two essential elements of digitization, which are internet access and mobile access. The data of this study has been prepared by the company "The Intelligence Internet Index" commissioned by "Facebook" and seeks to measure internet. The index assesses the performance of many countries in four categories: Accessibility, Affordability, Relevance and Readiness, and in our investigation, we will only take into account the following two elements of Accessibility (See Figures 3 and 4).

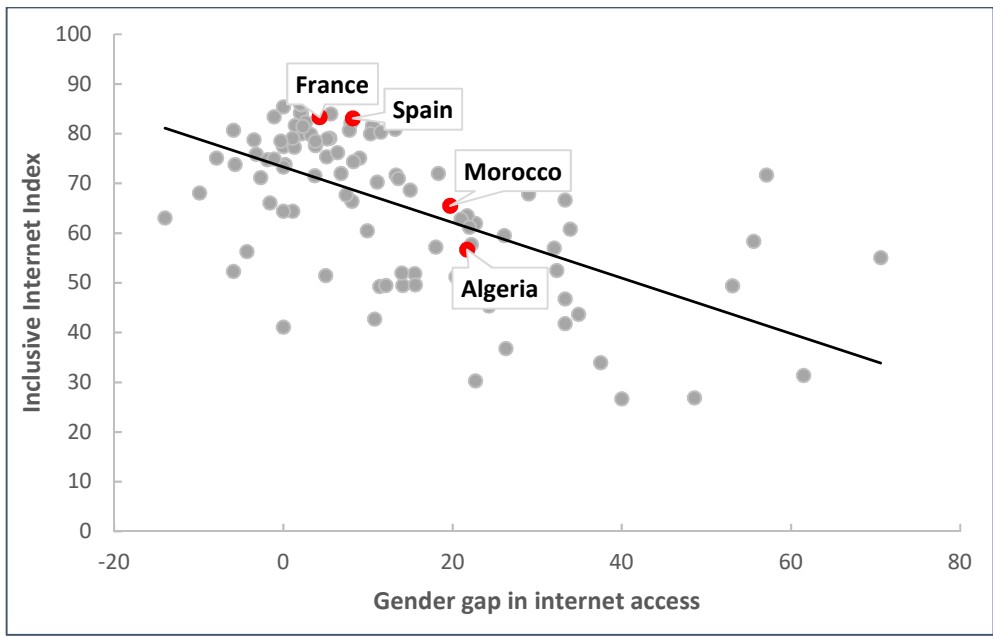

**Figure 3.** Relationship between internet access and gender (2019). (**Source**: Own elaboration from The Economist Intelligence Unit [100]).

Figure 3 clearly shows the difference between the four countries in terms of internet access according to gender. It is first observed that Algeria dominates this access gap, its gender fracture being higher than the others, followed by Morocco, which, according to the previous results, is more developed in terms of innovation than its Maghrebi neighbor but with greater gender inequalities. On the other hand, it is highlighted that Spain and France are positioned above the dividing line for having less gap in internet access and being part of the technological leaders.

The results are exactly the same in the case of access to the mobile telephone by continuation (Figure 4). We can observe that France and Spain occupy the same position among the leaders, unlike Morocco, which is a little over but always above the dividing line, and Algeria remains much more behind in this field.

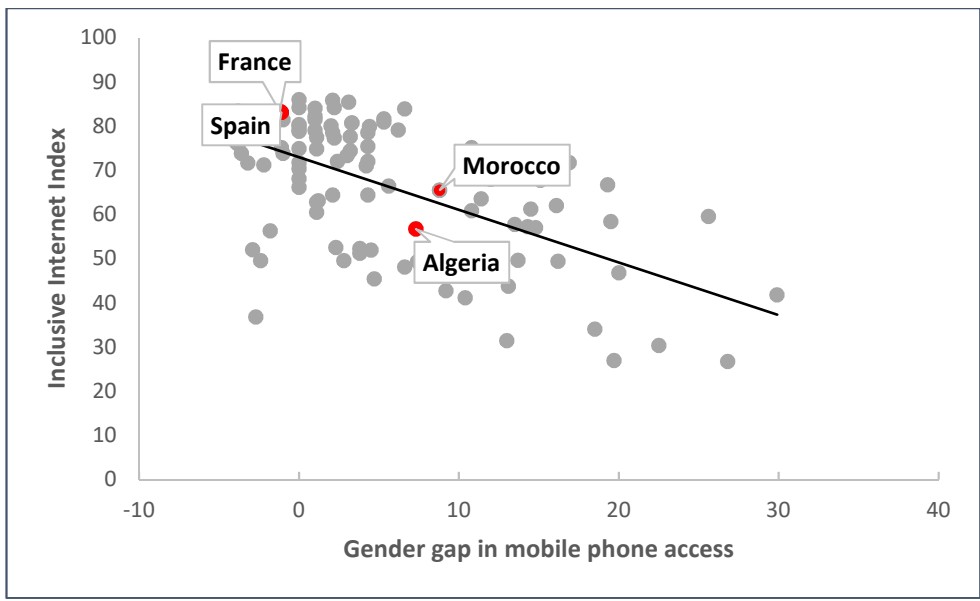

**Figure 4.** Relation between mobile access and gender (2019). (**Source:** Own elaboration from The Economist Intelligence Unit [100]).

## 4. Research Methodology and Empirical Analysis

To identify the influence of the Gender Digital Divide in achieving the Sustainable Development Goals (SDGs), a regression analysis was performed. This analysis subordinates the scope of the SDGs to the decrease in the distance between levels of access and use of ICTs between women and men, which translates, among other aspects, into the empowerment of women through access and ICT use.

### 4.1. Data Collection

The population studied is made up of the total number of countries in the world. In this context, it is based on information describing the progress of the SDGs [92] and data from the Inclusive Internet Index [100] offered globally. To carry out the proposed analysis and give more consistency to this investigation, a study frame was created with the information available for 87 countries.

### 4.2. Variables Definition

The definition of the factors that condition the achievement of the SDGs has become a critical aspect of carrying out this work, since no previous pieces of work that attempt to explain this process in terms of gender digital divide were found. This situation has meant that, in an exploratory way, a model which attempts to distinguish the relations inside the gender digital divide is proposed as an explanatory variable, controlling its effect with a series of variables that, due to their definition and global character, are considered as conditions widely meeting these objectives.

The independent variables included in the regression model are defined as (See Table 6):

- Gender Digital Divide: Distance between levels of ICT access between women and men. The resulting measure is a percentage that expresses values between 0 and 100, where 0 corresponds to total equality in the use of ICT and 100 corresponds to total inequality or the total empowerment of men in the use of ICT.
- Inclusive Internet Index: Index composed of four categories of information that indicate the degree of inclusiveness (Availability, Affordability, Relevance and Readiness) of all internet-related services in a country. The measure expresses a percentage between 0 and 100, where 0 corresponds to total exclusivity, and 100 corresponds to total inclusivity.
- Gini index: Measure of inequality in terms of wealth within a country. The measure expresses a percentage between 0 and 100, where 0 corresponds to total equality (everyone has the same wealth) and the value 100 corresponds to total inequality (inequality in the distribution of wealth).
- Gender divides: Distance between women and men made up of four categories of information indicating the degree of participation of women in economic participation and opportunity, for educational attainment, health and survival, and political empowerment, it should be remembered that this index does not include the technological aspect. The measure expresses a percentage between 0 and 100, where 0 corresponds to total equality and 100 corresponds to total inequality between men and women.

**Table 6.** Definitions of independent variables.

| Variable | Description | Mean/ Percent. | Sd. | Min. | Max. | Expected Effect |
|---|---|---|---|---|---|---|
| Gender digital divide | (0 = Total equality; 100 = Total inequality) | 9.77 | 12.33 | 0.00 | 69.40 | − |
| Inclusive Internet Index | (0 = Total exclusive; 100 = Total inclusive) | 66.96 | 15.14 | 26.70 | 86.00 | + |
| Gini | (0 = Total equality; 100 = Total inequality) | 38.64 | 7.42 | 26.50 | 63.00 | − |
| Gender divide | (0 =Total equality; 100 = Total inequality) | 29.55 | 5.54 | 17.80 | 45.00 | − |

**Source**: Produced by the author.

*4.3. Estimation Model*

We tackle the proposal of this article by estimating the following lineal regression model with ordinary least squared (OLS) procedure for each of the seven sustainable development goals (i.e., SDG2, SDG3, SDG4, SDG5, SDG8, SDG9 and SDG10) on a set of index.

$$SDG_{ji} = \beta_0 + \beta_1\,GDD_i + \beta_2 IINT_i + \beta_3 GINI_i + \beta_4 GD_i + \varepsilon_i \tag{1}$$

where:

$GDD$ = Gender Digital Divide Index.
$IINT$ = Inclusive Internet Index.
$GINI$ = Gini index.
$GD$ = Gender Divide Index.
$\varepsilon$ = Error term.
$j$ = Sustainable development goal in which we focused on ($j = 1 \dots 7$).

## 5. Results

For each of the seven sustainable development goals considered, firstly, we estimated the previous model without taking into account the GD variable (i.e., Gender Divide Index). Later, in order to make a sensitive analysis, we included that variable which gave us a global Gender Divide Index. In such a way, we can check if the estimated parameters of "Gender Digital Divide Index" change or not with its inclusion. The result obtained can be observed in Table 7.

**Table 7.** Results of the ordinary least squares regression analysis for the effects of the set of index on each SDG considered.

| Variables | SDG2 | | SDG3 | | SDG4 | | SDG5 | | SDG8 | | SDG9 | | SDG10 | |
|---|---|---|---|---|---|---|---|---|---|---|---|---|---|---|
| | *a* | *b* | *a* | *b* | *a* | *b* | *a* | *b* | *a* | *b* | *a* | *b* | *a* | *b* |
| GDD | −0.209 *** | −0.185 ** | −0.256 ** | −0.225 * | −0.312 ** | −0.254 * | −0.546 *** | −0.235 *** | −0.053 | 0.008 | −0.249 * | −0.167 | −0.334 ** | −0.236 |
| | (−2.74) | (−2.33) | (−2.36) | (−1.98) | (−2.33) | (−1.84) | (−3.92) | (−2.81) | (−0.70) | (0.11) | (−1.81) | (−1.19) | (−2.07) | (−1.38) |
| IINT | 0.226 *** | 0.228 *** | 0.796 *** | 0.797 *** | 0.873 *** | 0.877 *** | 0.065 | 0.084 | 0.288 *** | 0.292 *** | 1.16 *** | 1.16 *** | −0.282 * | −0.281 ** |
| | (3.43) | (3.45) | (5.45) | (8.47) | (7.54) | (7.61) | (0.54) | (1.20) | (4.40) | (4.64) | (9.75) | (9.97) | (−1.98) | (−2.00) |
| GINI | −0.307 *** | −0.314 *** | −0.458 *** | −0.466 *** | 0.054 | 0.038 | 0.136 | 0.050 | −0.214 ** | −0.230 ** | −0.433 ** | −0.456 ** | −2.79 *** | −2.82 *** |
| | (−2.90) | (−2.96) | (−3.03) | (−3.08) | (0.29) | (0.21) | (0.70) | (0.45) | (−2.03) | (−2.28) | (−2.27) | (−2.43) | (−12.29) | (−12.53) |
| GD | | −0.142 | | −0.190 | | −0.346 | | −1.885 *** | | −0.368 *** | | −0.496 ** | | −0.539 * |
| | | (−1.03) | | (−0.97) | | (−1.44) | | (−13.04) | | (−2.81) | | (−2.04) | | (−1.67) |
| F-value | 26.34 | 20.04 | 78.50 | 59.07 | 53.35 | 41.05 | 10.77 | 67.06 | 20.25 | 88.02 | 88.02 | 69.55 | 55.20 | 43.08 |
| (prob.) | (0.00) | (0.00) | (0.00) | (0.00) | (0.00) | (0.00) | (0.00) | (0.00) | (0.00) | (0.00) | (0.00) | (0.00) | (0.00) | (0.00) |
| $R^2$ | 0.48 | 0.49 | 0.74 | 0.74 | 0.66 | 0.67 | 0.28 | 0.76 | 0.42 | 0.76 | 0.76 | 0.77 | 0.69 | 0.70 |
| Adj. $R^2$ | 0.47 | 0.47 | 0.73 | 0.73 | 0.65 | 0.65 | 0.25 | 0.75 | 0.40 | 0.75 | 0.75 | 0.76 | 0.67 | 0.68 |

**Source**: Own elaboration; *** $p < 0.01$, ** $p < 0.05$, * $p < 0.10$; Number of observations = 87; A: Regression model without including GD; B: Regression model including GD.

As we can see for all the analyzed SDGs, the variables that have been included have the expected effect. In other words, as the "Gender Digital Divide" (GDD) increases, the proportion of the SDG that the country has reached decreases. For its part, an increase in the "Inclusive Internet" (IINT) index has a positive effect on this proportion in most of the SDGs analyzed.

Regarding the variable "Gender Divide" (GD), one can make two observations. On the one hand, it has a negative effect on the proportion of the SDG in question. On the other hand, it highlights the impact that its inclusion in the model has on the effect of the variable "Gender Digital Divide" (GDD) on the SDG in question (i.e., the parameter $\beta_1$). A reduction of this effect is observed when the variable "Gender Divide" (GD) is included in the models. In view of these results, we can say that, although the "Gender Digital Divide" has a negative effect on the achievement of the SDGs, there are other dimensions of the gender gap, other than the technological dimension, that further compromise the achievement of these objectives for the different countries.

## 6. Discussion and Conclusions

This analysis confirms that there is a complementarity between the development of information and telecommunications technologies, the gender equality and sustainable development.

In effect, sustainable development has the goal of combating inequalities, with responsible management of resources, an improvement in common wellbeing, plus guarantees of economic development.

ICTs are key, and an essential tool for social, economic and environmental transformations. Nurturing talent in the technology sector, without any exclusion, is therefore crucial in achieving sustainable development.

Right now, more efficient educational processes must be implemented that allow for the inclusion of everyone, and especially women, to enable their improved integration into the world of technology, as users, producers, researchers and entrepreneurs. In this sense, Wolfensohn [101] said: "If we educate a boy, we educate one person. If we educate a girl, we educate a family and a whole nation."

Indeed, the gender element is decisive in many of the SDGs, and this is because it allows doubling efforts and contributions in all sectors. As previously reported, women represent half of the population and incorporating them into development has become essential for all nations, and having them in the top management team means understanding different realities that exist on the market, whether it be consumers, clients, suppliers or other stakeholders [102].

It should be noted that the inclusion of women is not intended to replace that of men, but on the contrary, it aims to provide an enormous advantage to various sectors, and that is because they are complementary to men, and because they can respond to all demands of the new markets. To this effect, Slyke et al. [103] indicates that men and women differ in their use of information technology, such that women´s use intentions are more strongly influenced by the ease of use. On the other hand, Acevedo [104] confirms the existence of a gender complementarity that explains that the association of men (with strength, technique, authority and production orientation) and women (with communication, creativity, analysis and versatility) is a fact.

In this study, a big difference has been observed between the countries of the European Union and those of the Maghreb in terms of technology and gender indicators, and it has been detected how the northern Mediterranean countries are leading in these areas, which enables them to better achieve sustainable development goals. This is due to the country's resources and the degree of human development that are highlighted by the HDI and GDP in each country, and is justified by factors relating to each country's socio-political situation and also by their degree of economic development. This result tells us that interest in equality and in the technology sector has not been the same on each side of the Mediterranean. It has also been observed that none of the countries has achieved all the sustainable development goals, except for France, which has achieved goal 1, and that the rest of the SDGs present a more favorable evolution to achieve it in the year 2030, and that is thanks to the influence that ICTs and equality have in this country and other European countries.

On the other hand, from the investigation results, we can also highlight that the gender gap has a negative effect on the achievement of the analyzed sustainable development goals. When a bigger gap is noted, it is more difficult to achieve the objective. In that respect, Drumea et al. [105] argued that the gender inequality generates costs that have a negative social impact and lead to environmental degradation.

In the particular case of the digital gender gap, this negative effect also manifests itself, which justifies the importance of reducing said gap so that countries can achieve the proposed sustainable development goals.

Regarding the influence of ICTs' promotion on the proportion of the SDG that a country can achieve, the "Inclusive Internet" (IINT) index shows a positive effect. In this line, Fernández-Portillo et al. [70] argue that the deployment of ICTs can favor sustainable economic development, particularly when connectivity, human capital and use of internet grow. For its part, the GINI index measures wealth inequality, which is also directly related to the achievement of these objectives, as the results of the analysis show, when the wealth inequality is greater, the achievement of the sustainable development goals analyzed is compromised to a greater extent.

As a general conclusion, it could be mentioned that the technological advances and social progress lead to a liberation and opening of minds; women users of ICTs are currently in a minority, and this prevents them from fully enjoying the benefits of these technologies. This observed backwardness is due to several social, cultural and economic factors that can be summarized as follows:

➢ Women are more deeply affected by technological illiteracy than men. Without training, it is difficult to improve women's access to ICTs. This point is due to several phenomena, such as the assignment of roles and the voluntary or involuntary propagation of gender stereotypes that prevent women from aspiring to scientific and technical careers.

➢ Injustices in access to employment opportunities in the ICT sector, caused by horizontal (recruitment) and vertical (glass ceiling) segregation. Hernández et al. [106] support this theory, indicating that companies have to make an effort not only to achieve parity but also to try to achieve equality between women and men in working conditions, and adopt equality or other plans and innovative measures to promote equality.

The lack of training and education and the lack of respect for equal opportunities hinders both technological development and sustainable development; contrarily, it creates a fracture that leads to the exclusion of people, including women (discrimination by age, gender, etc.). In this sense, Linturi [107] says that individuals need to be educated, regulations and practices changed, and motivational rewards rethought if we wish society overall to benefit from technological progress.

Lastly, we need investment in infrastructures with more developed innovations that promote the fight against technological inequalities and provide new means and solutions, enabling the integration of women and contributing to work-life balance (e-work, for example).

The observed digital divide prevents some developing countries from accessing, using and enjoying ICTs in the same way as developed countries, and, in fact, prevents them from securing the same level of production and social participation.

In order to overcome this gender digital divide, policies must be put in place to guarantee gender equality in this field, and to raise awareness among all economic actors and civil society of the importance and value that the integration of women and their skills could contribute to the building of a fairer world.

In this sense, Castaño [35] indicates that sustainable progress must be merged on the basis of solidarity, and this entails not only sharing information, but also resources. Only in this way will it be possible to avoid the digital divide by having the advantages offered to our autonomy and empowerment by new information and communication technologies.

In short, having technological infrastructures and enjoying their use in an equal way allows countries to achieve several of the sustainable objectives and confirms our initial hypothesis.

**Author Contributions:** Conceptualization, H.K. and M.D.d.-M.G.; formal analysis, H.K. and M.D.d.-M.G.; investigation, H.K., J.L.S.-N. and E.I.L.-B.; data curation, H.K., J.L.S.-N. and E.I.L.-B.; writing—original draft preparation, H.K., J.L.S.-N., E.I.L.-B. and M.D.d.-M.G.; writing—review and editing, M.D.d.-M.G. and H.K.; supervision, M.D.d.-M.G. All authors have read and agreed to the published version of the manuscript.

**Funding:** The research did not receive any external funding.

**Acknowledgments:** The authors wish to thank the reviewers for all the contributions made to the enrichment of the work. H.K. acknowledges the UPCT´s support and opportunity and especially the Foundation of "Women in Affric" for the scholarship offered.

**Conflicts of Interest:** The authors declare no conflict of interest.

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
