# Peer review of "The Impact of the Gender Digital Divide on Sustainable Development: Comparative Analysis between the European Union and the Maghreb"

_sustainability, doi:10.3390/su12083347_

Round 1
Reviewer 1 Report
The paper addresses a topic of interest such as the impact of gender on sustainable development and has the potential to be published. However, some aspects must be improved:
1.- Sometimes the low participation of women in the technological sectors (lines 60-62; 240-247) or gender gap in the technology sector is confused with their role as users of new technologies or the potential they offer to reduce gender differences, where differences also persist, but where the potential for improvement in working conditions and quality of life is important.
2.- The paper focuses on 5 SDG (lines 156-158) pointing out that at first glance they seem the most related to the gender digital divide. This statement requires a greater justification, since in the opinion of this reviewer the relationship is not so obvious. Specifically, much has been written about the role of women in food security (SDG # 2), in improving rural family health (# 3) and the improvement of the nutritional security for their household, where ICTs have a great potential of. The focus on the previous objectives and not on the others , should be more justified
3.- More evidence should be provided of the relationship between the achievements and the evolution of the SDG indicators as the only element considered in the comparison between countries is the gender gap indicator.
Author Response
First, thank you for the comments on the first article version, which have been of great interest to us in reorienting our ideas.
The work refers to various elements that constitute the concept of digital divide, including access and use, the need for training and the role of women in the ICT sector. In fact, women are not only beneficiaries, but also protagonists. Therefore, the relationship between these two points has been further specified, indicating that women as ICT designers can influence the use of this tool, since she knows better which needs and difficulties women have to face and how they can be encouraged to use them.
The selection of the SDGs, supported by the literature review and the achievement results obtained from the analyzed countries, has been reviewed and analyzed through multiple linear regression. Based on the reviewer's indications, the two SDGs that were not initially planned have been included, as the reviewer has advised.
The results obtained from the application of the multiple linear regression method have also demonstrated the existence of a relationship between the digital gender gap and the achievement of the SDGs, as can be seen in Table 8.
We believe that this consideration is justified with the incorporation of a multiple linear regression analysis, incorporating two regression models, in which the relationship between the digital gender gap, the inclusive Internet index, the Gini index and the gender gap; this has determined, through the goodness of fit, the influence between these variables and the achievement of the selected SDGs. This justification has been added in red from line 657.
Reviewer 2 Report
The theme of the work is of interest, and faces a current and interesting topic, but does not have the quality required for a scientific article.
It has certain shortcomings in the introductory section, mainly due to the lack of bibliographic citations.
The literature review is not quality, because it relies too often on documents that are not published in scientific journals.
As for the methodology, it is far from being a suitable methodology for a research article, it is not presented, not even a simple regression, and the authors only rely on descriptive data to do their study. These types of methodologies are accepted in consulting reports, or end-of-study studies, but not in the academic world of scientific research because they are too poor to obtain conclusions.
The exposure of the results and conclusions are poor, but this is derived from the methodology used.
I encourage the authors, to continue working in this line, and to include a panel data analysis, in the case of time series analysis, and a multigroup analysis with PLS-SEM, in the case of individual data.
Author Response
We appreciate your feedback on the first version of the paper, which has encouraged us to improve it.
Following their recommendations, the introduction has been revised again and several citations have been provided to support the content of these. As a source of these citations, impact scientific journals, books and statistical data from official organizations (Like United Nations or ITU ) have been used. All these incorporations can be seen, in red, both in the introduction section, in the theoretical framework and in the References section.
Regarding the methodology, it has been reviewed and the analysis has been rethought in a different way, to attend to this consideration, the influence on the achievement of the selected SDGs of the digital gender gap and ICT has been studied, using a multiple linear regression. This has generated the incorporation of a new section, in nº 4, which defines the nature of the variables and formulation, completing the analysis in the discussion and conclusions section.
We thank you again for your indications, which has allowed us at this point to modify the presentation of results and conclusions, located on line 753.
The interest shown by the reviewer in this work is a question that encourages us to continue working in this regard, the methodologies indicated will be very taken into account in future work. We consider this study to be exploratory in nature, as we do not find in the existing literature works that relate the achievements of the SDGs to gender and ICT.
As already indicated, the multiple linear regression methodology has been used to analyze the influence of gender and ICT variables on the achievement of the SDGs.
Reviewer 3 Report
The idea for the paper is a good one. It does however need to be streamlined and a central more nuanced argument developed throughout the paper. The structure should be improved.
Regarding the introduction: The introduction should describe the main structure of the article. What are the main research questions that this article will try and answer, on what methods is the research based and how have the authors reached their conclusions. Why look at France, Spain, Morocco, Algeria? What is the justification for examining these four countries? It should introduce the conceptual framework (i.e. bringing together ICT and Sustainable Development). A minor issue in the introduction - the authors could make reference to the importance of women's involvement as technological 'designers', i.e. producers of technology and refer to including the gender dimension in design as 'gendered innovation'.
For section 3 How do ICTs influence sustainable development - a more critical assessment of ICTs is missing. For each 'Definition of SDGs' and their Link to ICTs' - a more critical assessment could also be made. For example, SDG 1- it could be argued that ICTs have exacerbated existing inequalities. SDG 5 for example a more critical approach could focus on the gendered nature of cyberbullying...also for SDG 7 and 13 the carbon footprint of the internet.. etc...
What is currently presented in this section is very one sided - this section should be more nuanced and evidence based.
The next section covers lots of different issues and does not seem to have a logical structure...
I think that the sentence "In effect there are various factors that restrict women's confidence in their own abilities to effectively perform in ICT related sectors, plus other factors that ultimately divert their attention from the possibility of enrolling in these technical careers." underestimates the more structural barriers.. Whilst these then go on to be described (i.e. vertical and horizontal segregation) - it is unclear the weight the authors give to the various explanations.......
In short I think the arguments need slimming down - choosing a couple of key concepts, bringing in the evidence base to construct the argument of the paper...
Author Response
Firstly, the authors wish to thank the reviewer for their comments on the first version of the paper, which have been of great interest to us in restructuring and modifying the article.
The recommendations regarding the introduction have been taken into consideration in their entirety and contributed in version 2. The selection of the countries has been justified, a brief comment has been placed at the end of the introduction, expanding from line 466, in order to carry out a comparison between the selected countries.
Although it has been considered of minor importance by the reviewer, however, we wish to add that the work refers to several elements that constitute the concept of digital divide, including access and use, training and the role of women in the ICT sector, that is, it is not only about women being beneficiaries, but also being protagonists. Therefore, the relationship between these two points has been further specified, indicating that women as ICT designers can influence the use of this tool, since she knows better what the needs and difficulties are that women have to face. women and how they can be encouraged to use them.
Regarding the influence of ICTs on sustainable development, it has been supported with bibliographic citations and specified in each point in version 2. The aspects highlighted by the reviewer of the different SDGs have been incorporated and extended comments. These contributions can be seen in Table1.
The methodology has also been reviewed and proposed with a scientific tool.
In version 2, it can be seen that the structure of the sections has been modified according to the reviewer's consideration.
The authors don´t pretend to underestimate structural barriers, on the contrary, they indicate that the lack of confidence of women makes them more easily accept the injustices related to horizontal and vertical segregation. For example: If women are faced with an injustice in terms of career advancement or training, the most likely reaction is that they will accept it, without asking for explanations, or complain close to the unions or labor inspection, and this is due to the fact that they first believe that they have less technical capabilities than men or for fear of not being up to their male colleagues and not being able to assume this responsibility as they do, since they have fewer family responsibilities and more time than to dedicate to the company than women.
We appreciate the synthesis of their contributions and we hope we have adequately reflected them in version 2.
Round 2
Reviewer 2 Report
The work has improved significantly, but still have room for improvement, in the literature review, and use of bibliography, where it is recommended to deepen the use of literature published in scientific journals of recognized quality, such as this same journal, and where they can find recent works that can serve as a reference to improve the literature review, for example this: https://www.mdpi.com/2071-1050/11/22/6307
Author Response
Thank you very much for the comments. We inform you that the literature review was done and indicated in blue color in the text. You can see 12 new references of impact journals in the new version of the article, including the one you proposed.
